# Doubly Reparameterized Gradient Estimators for Monte Carlo Objectives

**George Tucker**
Google Brain
gjt@google.com

**Dieterich Lawson**
New York University
jdl404@nyu.edu

**Shixiang Gu**
Google Brain
shanegu@google.com

**Chris J. Maddison**
University of Oxford, DeepMind
cmaddis@stats.ox.ac.uk

## Abstract

Deep latent variable models have become a popular model choice due to the scalable learning algorithms introduced by (Kingma & Welling, 2013; Rezende et al., 2014). These approaches maximize a variational lower bound on the intractable log likelihood of the observed data. Burda et al. (2015) introduced a multi-sample variational bound, IWAE, that is at least as tight as the standard variational lower bound and becomes increasingly tight as the number of samples increases. Counterintuitively, the typical inference network gradient estimator for the IWAE bound performs poorly as the number of samples increases (Rainforth et al., 2018; Le et al., 2018). Roeder et al. (2017) propose an improved gradient estimator, however, are unable to show it is unbiased. We show that it is in fact biased and that the bias can be estimated efficiently with a second application of the reparameterization trick. The *doubly reparameterized gradient* (DReG) estimator does not suffer as the number of samples increases, resolving the previously raised issues. The same idea can be used to improve many recently introduced training techniques for latent variable models. In particular, we show that this estimator reduces the variance of the IWAE gradient, the reweighted wake-sleep update (RWS) (Bornschein & Bengio, 2014), and the jackknife variational inference (JVI) gradient (Nowozin, 2018). Finally, we show that this computationally efficient, unbiased drop-in gradient estimator translates to improved performance for all three objectives on several modeling tasks.

## 1 Introduction

Following the influential work by (Kingma & Welling, 2013; Rezende et al., 2014), deep generative models with latent variables have been widely used to model data such as natural images (Rezende & Mohamed, 2015; Kingma et al., 2016; Chen et al., 2016; Gulrajani et al., 2016), speech and music time-series (Chung et al., 2015; Fraccaro et al., 2016; Krishnan et al., 2015), and video (Babaeizadeh et al., 2017; Ha & Schmidhuber, 2018; Denton & Fergus, 2018). The power of these models lies in combining learned nonlinear function approximators with a principled probabilistic approach, resulting in expressive models that can capture complex distributions. Unfortunately, the nonlinearities that empower these model also make marginalizing the latent variables intractable, rendering direct maximum likelihood training inapplicable. Instead of directly maximizing the marginal likelihood, a common approach is to maximize a tractable lower bound on the likelihood such as the variational evidence lower bound (ELBO) (Jordan et al., 1999; Blei et al., 2017). The tightness of the bound is determined by the expressiveness of the variational family. For tractability, a factorized variational family is commonly used, which can cause the learned model to be overly simplistic.

Burda et al. (2015) introduced a multi-sample bound, IWAE, that is at least as tight as the ELBO and becomes increasingly tight as the number of samples increases. Counterintuitively, although

---

Implementation of DReG estimators and code to reproduce experiments: sites.google.com/view/dregs.

the bound is tighter, Rainforth et al. (2018) theoretically and empirically showed that the standard inference network gradient estimator for the IWAE bound performs poorly as the number of samples increases due to a diminishing signal-to-noise ratio (SNR). This motivates the search for novel gradient estimators.

Roeder et al. (2017) proposed a lower-variance estimator of the gradient of the IWAE bound. They speculated that their estimator was unbiased, however, were unable to prove the claim. We show that it is in fact biased, but that it is possible to construct an unbiased estimator with a second application of the reparameterization trick which we call the IWAE *doubly reparameterized gradient* (DReG) estimator. Our estimator is an unbiased, computationally efficient drop-in replacement, and does not suffer as the number of samples increases, resolving the counterintuitive behavior from previous work (Rainforth et al., 2018). Furthermore, our insight is applicable to alternative multi-sample training techniques for latent variable models: reweighted wake-sleep (RWS) (Bornschein & Bengio, 2014) and jackknife variational inference (JVI) (Nowozin, 2018).

In this work, we derive DReG estimators for IWAE, RWS, and JVI and demonstrate improved scaling with the number of samples on a simple example. Then, we evaluate DReG estimators on MNIST generative modeling, Omniglot generative modeling, and MNIST structured prediction tasks. In all cases, we demonstrate substantial unbiased variance reduction, which translates to improved performance over the original estimators.

## 2 BACKGROUND

Our goal is to learn a latent variable generative model $p_\theta(x, z) = p_\theta(z)p_\theta(x|z)$ where $x$ are observed data and $z$ are continuous latent variables. The marginal likelihood of the observed data, $p_\theta(x) = \int p_\theta(x, z)\, dz$, is generally intractable. Instead, we maximize a variational lower bound on $\log p_\theta(x)$ such as the ELBO

$$\log p_\theta(x) = \log \mathbb{E}_{p_\theta(z)}[p_\theta(x|z)] \geq \mathbb{E}_{q(z|x)}\left[\log \frac{p_\theta(x, z)}{q(z|x)}\right], \tag{1}$$

where $q(z|x)$ is a variational distribution. Following the influential work by (Kingma & Welling, 2013; Rezende et al., 2014), we consider the amortized inference setting where $q_\phi(z|x)$, referred to as the inference network, is a learnable function parameterized by $\phi$ that maps from $x$ to a distribution over $z$. The tightness of the bound is coupled to the expressiveness of the variational family (i.e., $\{q_\phi\}_\phi$). As a result, limited expressivity of $\{q_\phi\}_\phi$, can negatively affect the learned model.

Burda et al. (2015) introduced the importance weighted autoencoder (IWAE) bound which alleviates this coupling

$$\mathbb{E}_{z_{1:K}}\left[\log\left(\frac{1}{K}\sum_{i=1}^{K}\frac{p_\theta(x, z_i)}{q_\phi(z_i|x)}\right)\right] \leq \log p_\theta(x), \tag{2}$$

with $z_{1:K} \sim \prod_i q_\phi(z_i|x)$. The IWAE bound reduces to the ELBO when $K = 1$, is non-decreasing as $K$ increases, and converges to $\log p_\theta(x)$ as $K \to \infty$ under mild conditions (Burda et al., 2015). When $q_\phi$ is reparameterizable[1], the standard gradient estimator of the IWAE bound is

$$\nabla_{\theta,\phi}\mathbb{E}_{z_{1:K}}\left[\log\left(\frac{1}{K}\sum_{i=1}^{K} w_i\right)\right] = \nabla_{\theta,\phi}\mathbb{E}_{\epsilon_{1:K}}\left[\log\left(\frac{1}{K}\sum_{i=1}^{K} w_i\right)\right] = \mathbb{E}_{\epsilon_{1:K}}\left[\sum_{i=1}^{K}\frac{w_i}{\sum_j w_j}\nabla_{\theta,\phi}\log w_i\right]$$

where $w_i = p_\theta(x, z_i)/q_\phi(z_i|x)$. A single sample estimator of this expectation is typically used as the gradient estimator.

As $K$ increases, the bound becomes increasingly tight, however, Rainforth et al. (2018) show that the signal-to-noise ratio (SNR) of the inference network gradient estimator goes to $0$. This does not

---

[1]Meaning that we can express $z_i$ as $z(\epsilon_i, \phi)$, where $z$ is a deterministic, differentiable function and $p(\epsilon_i)$ does not depend on $\theta$ or $\phi$. This allows gradients to be estimated using the reparameterization trick (Kingma & Welling, 2013; Rezende et al., 2014). Notably, the recently introduced implicit reparameterization trick has expanded the class of distributions for which we can compute reparameterization gradients to include Gaussian mixtures among many others (Graves, 2016; Jankowiak & Obermeyer, 2018; Jankowiak & Karaletsos, 2018; Figurnov et al., 2018).

happen for the model parameters ($\theta$). Following up on this work, Le et al. (2018) demonstrate that this deteriorates the performance of learned models on practical problems.

Because the IWAE bound converges to $\log p_\theta(x)$ (as $K \to \infty$) regardless of $q_\phi$, $\phi$'s affect on the bound must diminish as $K$ increases. It may be tempting to conclude that the SNR of the inference network gradient estimator must also decrease as $K \to \infty$. However, low SNR is a limitation of the gradient estimator, not necessarily of the bound. Although the magnitude of the gradient converges to 0, if the variance of the gradient estimator decreases more quickly, then the SNR of the gradient estimator need not degrade. This motivates the search for lower variance inference network gradient estimators.

To derive improved gradient estimators for $\phi$, it is informative to expand the total derivative[2] of the IWAE bound with respect to $\phi$

$$\mathbb{E}_{\epsilon_{1:K}} \left[ \sum_{i=1}^{K} \frac{w_i}{\sum_{j=1}^{K} w_j} \left( -\frac{\partial}{\partial \phi} \log q_\phi(z_i|x) + \frac{\partial \log w_i}{\partial z_i} \frac{dz_i}{d\phi} \right) \right]. \tag{3}$$

Previously, Roeder et al. (2017) found that the first term within the parentheses of Eq. 3 can contribute significant variance to the gradient estimator. When $K = 1$, this term analytically vanishes in expectation, so when $K > 1$ they suggested dropping it. Below, we abbreviate this estimator as STL. As we show in Section 6.1, the STL estimator introduces bias when $K > 1$.

## 3 DOUBLY REPARAMETERIZED GRADIENT ESTIMATORS (DREGS)

Our insight is that we can estimate the first term within the parentheses of Eq. 3 efficiently with a second application of the reparameterization trick. To see this, first note that

$$\mathbb{E}_{\epsilon_{1:K}} \left[ \sum_{i=1}^{K} \frac{w_i}{\sum_{j=1}^{K} w_j} \frac{\partial}{\partial \phi} \log q(z_i|x) \right] = \sum_{i=1}^{K} \mathbb{E}_{\epsilon_{1:K}} \left[ \frac{w_i}{\sum_{j=1}^{K} w_j} \frac{\partial}{\partial \phi} \log q(z_i|x) \right],$$

so it suffices to focus on one of the $K$ terms. Because the derivative is a partial derivative $\frac{\partial}{\partial \phi}$, it treats $z_i = z(\epsilon_i, \phi)$ as a constant, so we can freely change the random variable that the expectation is over to $z_{1:K}$. Now,

$$\mathbb{E}_{z_{1:K}} \left[ \frac{w_i}{\sum_j w_j} \frac{\partial}{\partial \phi} \log q_\phi(z_i|x) \right] = \mathbb{E}_{z_{-i}} \mathbb{E}_{z_i} \left[ \frac{w_i}{\sum_j w_j} \frac{\partial}{\partial \phi} \log q_\phi(z_i|x) \right], \tag{4}$$

where $z_{-i} = z_{1:i-1,i+1:K}$ is the set of $z_{1:K}$ without $z_i$. The inner expectation resembles a REINFORCE gradient term (Williams, 1992), where we interpret $\frac{w_i}{\sum_j w_j}$ as the "reward". Now, we can use the following well-known equivalence between the REINFORCE gradient and the reparameterization trick gradient (See Appendix 8.1 for a derivation)

$$\mathbb{E}_{q_\phi(z|x)} \left[ f(z) \frac{\partial}{\partial \phi} \log q_\phi(z|x) \right] = \mathbb{E}_\epsilon \left[ \frac{\partial f(z)}{\partial z} \frac{\partial z(\epsilon, \phi)}{\partial \phi} \right]. \tag{5}$$

This holds even when $f$ depends on $\phi$. Typically, the reparameterization gradient estimator has lower variance than the REINFORCE gradient estimator because it directly takes advantage of the derivative of $f$. Applying the identity from Eq. 5 to the right hand side of Eq. 4 gives

$$\mathbb{E}_{z_i} \left[ \frac{w_i}{\sum_j w_j} \frac{\partial}{\partial \phi} \log q_\phi(z_i|x) \right] = \mathbb{E}_{\epsilon_i} \left[ \frac{\partial}{\partial z_i} \left( \frac{w_i}{\sum_j w_j} \right) \frac{\partial z_i}{\partial \phi} \right]$$

$$= \mathbb{E}_{\epsilon_i} \left[ \left( \frac{1}{\sum_j w_j} - \frac{w_i}{(\sum_j w_j)^2} \right) \frac{\partial w_i}{\partial z_i} \frac{\partial z_i}{\partial \phi} \right] = \mathbb{E}_{\epsilon_i} \left[ \left( \frac{w_i}{\sum_j w_j} - \frac{w_i^2}{(\sum_j w_j)^2} \right) \frac{\partial \log w_i}{\partial z_i} \frac{\partial z_i}{\partial \phi} \right]. \tag{6}$$

This last expression can be efficiently estimated with a single Monte Carlo sample. When $z_i$ is not reparameterizable (e.g., the models in (Mnih & Rezende, 2016)), we can use a control variate (e.g.,

---

[2]$\log w_i$ depends on $\phi$ in two ways: $\phi$ and $z_i = z(\epsilon_i, \phi)$. The total derivative accounts for both sources of dependence and the partial derivative $\frac{\partial}{\partial \phi}$ considers $z_i$ as a constant.

$\frac{1}{K}\frac{\partial}{\partial\phi}\log q_\phi(z_i|x)$). In both cases, when $K = 1$, this term vanishes exactly and we recover the estimator proposed in (Roeder et al., 2017) for the ELBO. However, when $K > 1$, there is no reason to believe this term will analytically vanish.

Substituting Eq. 6 into Eq. 3, we obtain a simplification due to cancellation of terms

$$\nabla_\phi \mathbb{E}_{z_{1:K}}\left[\log\left(\frac{1}{K}\sum_{i=1}^K w_i\right)\right] = \mathbb{E}_{\epsilon_{1:K}}\left[\sum_{i=1}^K \left(\frac{w_i}{\sum_j w_j}\right)^2 \frac{\partial \log w_i}{\partial z_i}\frac{\partial z_i}{\partial \phi}\right]. \tag{7}$$

We call the algorithm that uses the single sample Monte Carlo estimator of this expression for the inference network gradient the IWAE doubly reparameterized gradient estimator (IWAE-DReG). This estimator has the property that when $q(z|x)$ is optimal (i.e., $q(z|x) = p(z|x)$), the estimator vanishes exactly and has zero variance, whereas this does not hold for the standard IWAE gradient estimator. We provide an asymptotic analysis of the IWAE-DReG estimator in Appendix 8.2. The conclusion of that analysis is that, in contrast to the standard IWAE gradient estimator, the SNR of the IWAE-DReG estimator exhibits the same scaling behaviour of $\mathcal{O}(\sqrt{K})$ for both the generation and inference network gradients (i.e., improving in $K$).

## 4 ALTERNATIVE TRAINING ALGORITHMS

Now, we review alternative training algorithms for deep generative models and derive their doubly reparameterized versions.

### 4.1 REWEIGHTED WAKE SLEEP (RWS)

Bornschein & Bengio (2014) introduced RWS, an alternative multi-sample update for latent variable models that uses importance sampling. Computing the gradient of the log marginal likelihood

$$\nabla_\theta \log p_\theta(x) = \frac{\nabla_\theta \int_z p_\theta(x, z)\, dz}{p_\theta(x)} = \frac{\int_z p_\theta(x, z)\nabla_\theta \log p_\theta(x, z)\, dz}{p_\theta(x)} = \mathbb{E}_{p_\theta(z|x)}\left[\nabla_\theta \log p_\theta(x, z)\right],$$

requires samples from $p_\theta(z|x)$, which is generally intractable. We can approximate the gradient with a self-normalized importance sampling estimator

$$\mathbb{E}_{p_\theta(z|x)}\left[\nabla_\theta \log p_\theta(x, z)\right] \approx \mathbb{E}_{z_{1:K}}\left[\sum_{i=1}^K \frac{w_i}{\sum_j w_j}\nabla_\theta \log p_\theta(x, z_i)\right],$$

where $z_{1:K} \sim \prod_i q_\phi(z_i|x)$. Interestingly, this is precisely the same as the IWAE gradient of $\theta$, so the RWS update for $\theta$ can be interpreted as maximizing the IWAE lower bound in terms of $\theta$. Instead of optimizing a joint objective for $p$ and $q$, RWS optimizes a separate objective for the inference network. (Bornschein & Bengio, 2014) propose a "wake" update and a "sleep" update for the inference network. Le et al. (2018) provide empirical support for solely using the wake update for the inference network, so we focus on that update.

The wake update approximately minimizes the KL divergence from $p_\theta(z|x)$ to $q_\phi(z|x)$. The gradient of the KL term is

$$\nabla_\phi \mathbb{E}_{p_\theta(z|x)}\left[\log p_\theta(z|x) - \log q_\phi(z|x)\right] = -\mathbb{E}_{p_\theta(z|x)}\left[\frac{\partial}{\partial \phi}\log q_\phi(z|x)\right].$$

The wake update of the inference network approximates the intractable expectation by self-normalized importance sampling

$$-\mathbb{E}_{p_\theta(z|x)}\left[\frac{\partial}{\partial \phi}\log q_\phi(z|x)\right] \approx -\mathbb{E}_{z_{1:K}}\left[\sum_{i=1}^K \frac{w_i}{\sum_j w_j}\frac{\partial}{\partial \phi}\log q_\phi(z_i|x)\right], \tag{8}$$

with $z_i \sim q_\phi(z_i|x)$. Le et al. (2018) note that this update does not suffer from diminishing SNR as $K$ increases. However, a downside is that the updates for $p$ and $q$ are not gradients of a unified objective, so could potentially lead to instability or divergence.

DOUBLY REPARAMETERIZED REWEIGHTED WAKE UPDATE

The wake update gradient for the inference network (Eq. 8) can be reparameterized

$$- \mathbb{E}_{z_{1:K}} \left[ \sum_{i=1}^{K} \frac{w_i}{\sum_j w_j} \frac{\partial}{\partial \phi} \log q_\phi(z_i|x) \right] = \mathbb{E}_{\epsilon_{1:K}} \left[ \sum_{i=1}^{K} \left( \frac{w_i^2}{(\sum_j w_j)^2} - \frac{w_i}{\sum_j w_j} \right) \frac{\partial \log w_i}{\partial z_i} \frac{\partial z_i}{\partial \phi} \right].$$
(9)

We call the algorithm that uses the single sample Monte Carlo estimator of this expression as the wake update for the inference network RWS-DReG.

Interestingly, the inference network gradient estimator from (Roeder et al., 2017) can be seen as the sum of the IWAE gradient estimator and the wake update of the inference network (as the wake update minimizes, we add the negative of Eq. 9). Their positive results motivate further exploration of convex combinations of IWAE-DReG and RWS-DReG

$$\mathbb{E}_{\epsilon_{1:K}} \left[ \sum_{i=1}^{K} \left( \alpha \frac{w_i}{\sum_j w_j} + (1 - 2\alpha) \frac{w_i^2}{(\sum_j w_j)^2} \right) \frac{\partial \log w_i}{\partial z_i} \frac{\partial z_i}{\partial \phi} \right].$$
(10)

We refer to the algorithm that uses the single sample Monte Carlo estimator of this expression as DReG($\alpha$). When $\alpha = 1$, this reduces to RWS-DReG, when $\alpha = 0$, this reduces to IWAE-DReG and when $\alpha = 0.5$, this reduces STL.

## 4.2 JACKKNIFE VARIATIONAL INFERENCE (JVI)

Alternatively, Nowozin (2018) reinterprets the IWAE lower bound as a biased estimator for the log marginal likelihood. He analyzes the bias and introduces a novel family of estimators, Jackknife Variational Inference (JVI), which trade off reduction in bias for increased variance. This additional flexibility comes at the cost of no longer being a stochastic lower bound on the log marginal likelihood. The first-order JVI has significantly reduced bias compared to IWAE, which empirically results in a better estimate of the log marginal likelihood with fewer samples (Nowozin, 2018). For simplicity, we focus on the first-order JVI estimator

$$K \times \mathbb{E}_{z_{1:K}} \left[ \log \left( \frac{1}{K} \sum_{i=1}^{K} w_i \right) \right] - \frac{K-1}{K} \sum_{i=1}^{K} \mathbb{E}_{z_{-i}} \left[ \log \left( \frac{1}{K-1} \sum_{j \neq i} w_j \right) \right].$$

It is straightforward to apply our approach to higher order JVI estimators.

DOUBLY REPARAMETERIZED JACKKNIFE VARIATIONAL INFERENCE (JVI)

The JVI estimator is a linear combination of $K$ and $K - 1$ sample IWAE estimators, so we can use the doubly reparameterized gradient estimator (Eq. 7) for each term.

## 5 RELATED WORK

Mnih & Rezende (2016) introduced a generalized framework of Monte Carlo objectives (MCO). The log of an unbiased marginal likelihood estimator is a lower bound on the log marginal likelihood by Jensen's inequality. In this view, the ELBO can be seen as the MCO corresponding to a single importance sample estimator of the marginal likelihood with $q_\theta$ as the proposal distribution. Similarly, IWAE corresponds to the $K$-sample estimator. Maddison et al. (2017) show that the tightness of an MCO is directly related to the variance of the underlying estimator of the marginal likelihood.

However, Rainforth et al. (2018) point out issues with gradient estimators of multi-sample lower bounds. In particular, they show that although the IWAE bound is tighter, the standard IWAE gradient estimator's SNR scales poorly with large numbers of samples, leading to degraded performance. Le et al. (2018) experimentally investigate this phenomenon and provide empirical evidence of this degradation across multiple tasks. They find that RWS (Bornschein & Bengio, 2014) does not suffer from this issue and find that it can outperform models trained with the IWAE bound. We conclude that it is not sufficient to just tighten the bound; it is important to understand the gradient estimators of the tighter bound as well.

Wake-sleep is an alternative approach to fitting deep generative models, first introduced in (Hinton et al., 1995) as a method for training Hemholtz machines. It was extended to the multi-sample setting by (Bornschein & Bengio, 2014) and the sequential setting in (Gu et al., 2015). It has been applied to generative modeling of images (Ba et al., 2015).

## 6 EXPERIMENTS

To evaluate DReG estimators, we first measure variance and signal-to-noise ratio (SNR)[3] of gradient estimators on a toy example which we can carefully control. Then, we evaluate gradient variance and model learning on MNIST generative modeling, Omniglot generative modeling, and MNIST structured prediction tasks.

### 6.1 TOY GAUSSIAN

We reimplemented the Gaussian example from (Rainforth et al., 2018). Consider the generative model with $z \sim N(\theta, I)$ and $x|z \sim N(z, I)$ and inference network $q_\phi(z|x) \sim N(Ax + b, \frac{2}{3}I)$, where $\phi = \{A, b\}$. As in (Rainforth et al., 2018), we sample a set of parameters for the model and inference network close to the optimal parameters (perturbed by zero-mean Gaussian noise with standard deviation 0.01), then estimate the gradient of the inference network parameters for increasing number of samples ($K$).

In addition to signal-to-noise ratio (SNR), we plot the squared bias and variance of the gradient estimators[4] in Fig. 1. The bias is computed relative to the expected value of the IWAE gradient estimator. As a result, although the average of $K$ ELBO gradient estimators is an unbiased estimator of the ELBO gradient, it is a *biased* gradient estimator of the IWAE objective. Importantly, SNR does not penalize estimators that are biased, so trivial constant estimators can have infinite SNR. Thus, it is important to consider additional evaluation measures as well. As $K$ increases, the SNR of the IWAE-DReG estimator increases, whereas the SNR of the standard gradient estimator of IWAE goes to 0, as previously reported. Furthermore, we can see the bias present in the STL estimator. As a check of our implementation, we verified that the observed "bias" for IWAE-DReG was statistically indistinguishable from 0 with a paired t-test. For the biased estimators (e.g., STL), we could easily reject the null hypothesis with few samples.

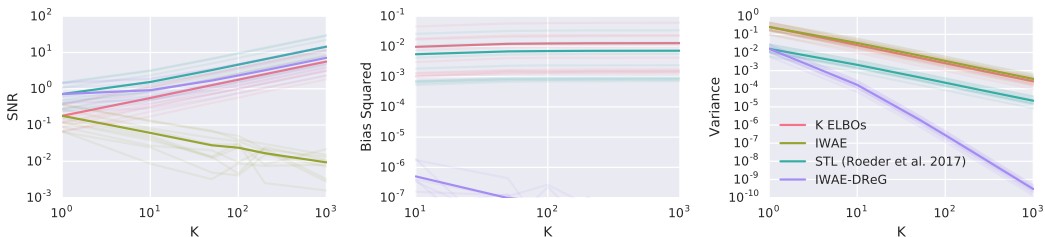

Figure 1: Signal-to-noise ratios (SNR), bias squared, and variance of gradient estimators with increasing $K$ over 10 random trials with 1000 measurement samples per trial (mean in bold). The observed "bias" for IWAE-DReG is not statistically significant under a paired t-test (as expected because IWAE-DReG is unbiased). IWAE-DReG is unbiased, its SNR increases with K, and it has the lowest variance of the estimators considered here.

### 6.2 GENERATIVE MODELING

Training generative models of the binarized MNIST digits dataset is a standard benchmark task for latent variable models. For this evaluation, we used the single latent layer architecture from (Burda et al., 2015). The generative model used 50 Gaussian latent variables with an isotropic prior and

---

[3]Defined as the mean of the estimator divided by the standard deviation.
[4]All dimensions of $\phi$ behaved qualitatively similarly, so for clarity, we show curves for a single randomly chosen dimension of $\phi$.

passed $z$ through two deterministic layers of 200 tanh units to parameterize factorized Bernoulli outputs. The inference network passed $x$ through two deterministic layers of 200 tanh units to parameterize a factorized Gaussian distribution over $z$. Because our interest was in improved gradient estimators and optimization performance, we used the dynamically binarized MNIST dataset, which minimally suffers from overfitting. We used the standard split of MNIST into train, validation, and test sets.

We trained models with the IWAE gradient, the RWS wake update, and with the JVI estimator. In all three cases, the doubly reparameterized gradient estimator reduced variance[5] and as a result substantially improved performance (Fig. 2).

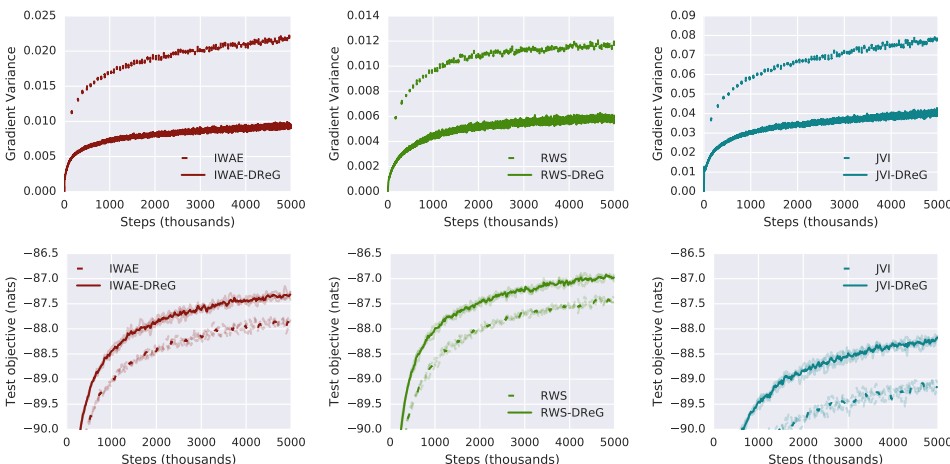

Figure 2: MNIST generative modeling trained according to IWAE (left), RWS (middle), and JVI (right). The top row compares the variance of the original gradient estimator (dashed) with the variance of the doubly reparameterized gradient estimator (solid). The bottom row compares test performance. The left and middle plots show the IWAE (stochastic) lower bound on the test set. The right plot shows the JVI estimator (which is not a bound) on the test set. The bold lines are the average over three trials, and individual trials are displayed as semi-transparent). All methods used $K = 64$.

We found similar behavior with different numbers of samples (Fig. 3 and Appendix Fig. 8). Interestingly, the biased gradient estimators STL and RWS-DReG perform best on this task with RWS-DReG slightly outperforming STL. As observed in (Le et al., 2018), RWS increasingly outperforms IWAE as $K$ increases. Finally, we experimented with convex combinations of IWAE-DReG and RWS-DReG (right Fig. 3). On this dataset, convex combinations that heavily weighted RWS-DReG had the best performance. However, as we show below, this is task dependent.

Next, we performed the analogous experiment with the dynamically binarized Omniglot dataset using the same model architecture. Again, we found that the doubly reparameterized gradient estimator reduced variance and as a result improved test performance (Figs. 5 and 6 in the Appendix).

## 6.3 STRUCTURED PREDICTION ON MNIST

Structured prediction is another common benchmark task for latent variable models. In this task, our goal is to model a complex observation $x$ given a context $c$ (i.e., model the conditional distribution $p(x|c)$). We can use a conditional latent variable model $p_\theta(x, z|c) = p_\theta(x|z, c)p_\theta(z|c)$, however, as before, computing the marginal likelihood is generally intractable. It is straightforward to adapt the bounds and techniques from the previous section to this problem.

---

[5]We summarized the Covariance matrix of the gradient estimators by reporting the trace of the Covariance normalized by the number of parameters. To estimate this quantity, we estimated moments from exponential moving averages with decay=0.99 (we found that the results were robust to the exact value of the decay).

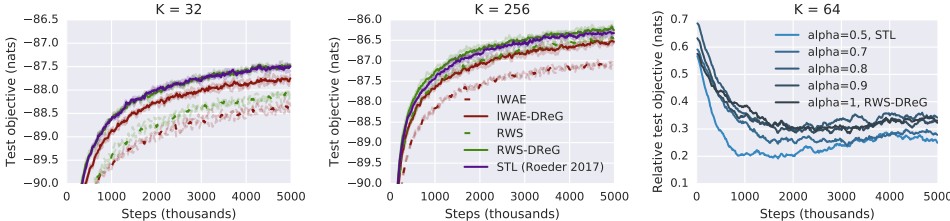

Figure 3: Log-likelihood lower bounds for generative modeling on MNIST. The left and middle plots compare performance with different number of samples $K = 32, 256$. For clarity the legend is shared between the plots. The bold lines are the average over three trials, and individual trials are displayed as semi-transparent). The right plot compares performance as the convex combination between IWAE-DReG and RWS-DReG is varied (Eq. 10). To highlight differences, we plot the difference between the test IWAE bound and the test IWAE bound IWAE-DReG achieved at that step.

To evaluate our method in this context, we use the standard task of modeling the bottom half of a binarized MNIST digit from the top half. We use a similar architecture, but now learn a conditional prior distribution $p_\theta(z|c)$ where $c$ is the top half of the MNIST digit. The conditional prior feeds $c$ to two deterministic layers of 200 tanh units to parameterize a factorized Gaussian distribution over $z$. To model the conditional distribution $p_\theta(x|c,z)$, we concatenate $z$ with $c$ and feed it to two deterministic layers of 200 tanh units to parameterize factorized Bernoulli outputs.

As in the previous tasks, the doubly reparameterized gradient estimator improves across all three updates (IWAE, RWS, and JVI; Appendix Fig. 7). However, on this task, the biased estimators (STL and RWS) underperform unbiased IWAE gradient estimators (Fig. 4). In particular, RWS becomes unstable later in training. We suspect that this is because RWS does not directly optimize a consistent objective.

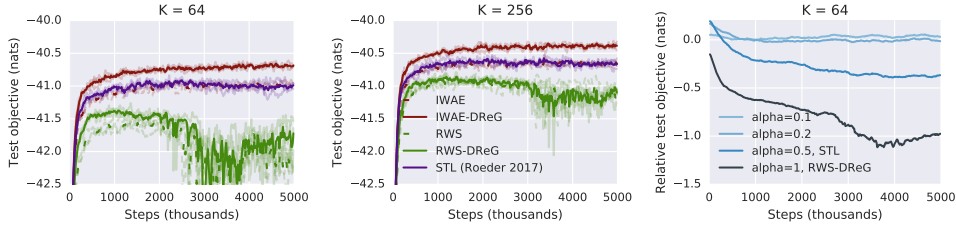

Figure 4: Log-likelihood lower bounds for structured prediction on MNIST. The left plot uses $K = 64$ samples and the right plot uses $K = 256$ samples. For clarity the legend is shared between the plots. The bold lines are the average over three trials, and individual trials are displayed as semi-transparent). The right plot compares performance as the convex combination between IWAE-DReG and RWS-DReG is varied (Eq. 10). To highlight differences, we plot the difference between the test IWAE bound and the test IWAE bound IWAE-DReG achieved at that step.

## 7 DISCUSSION

In this work, we introduce doubly reparameterized estimators for the updates in IWAE, RWS, and JVI. We demonstrate that across tasks they provide unbiased variance reduction, which leads to improved performance. Furthermore, DReG estimators have the same computational cost as the original estimators. As a result, we recommend that DReG estimators be used instead of the typical gradient estimators.

Variational Sequential Monte Carlo (Maddison et al., 2017; Naesseth et al., 2018; Le et al., 2018) and Neural Adapative Sequential Monte Carlo (Gu et al., 2015) extend IWAE and RWS to sequential

latent variable models, respectively. It would be interesting to develop DReG estimators for these approaches as well.

We found that a convex combination of IWAE-DReG and RWS-DReG performed best, however, the weighting was task dependent. In future work, we intend to apply ideas from (Baydin et al., 2017) to automatically adapt the weighting based on the data.

Finally, the form of the IWAE-DReG estimator (Eq. 7) is surprisingly simple and suggests that there may be a more direct derivation that is applicable to general MCOs.

## ACKNOWLEDGMENTS

We thank Ben Poole and Diederik P. Kingma for helpful discussion and comments on drafts of this paper. We thank Sergey Levine and Jascha Sohl-Dickstein for insightful discussion.

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

# 8 APPENDIX

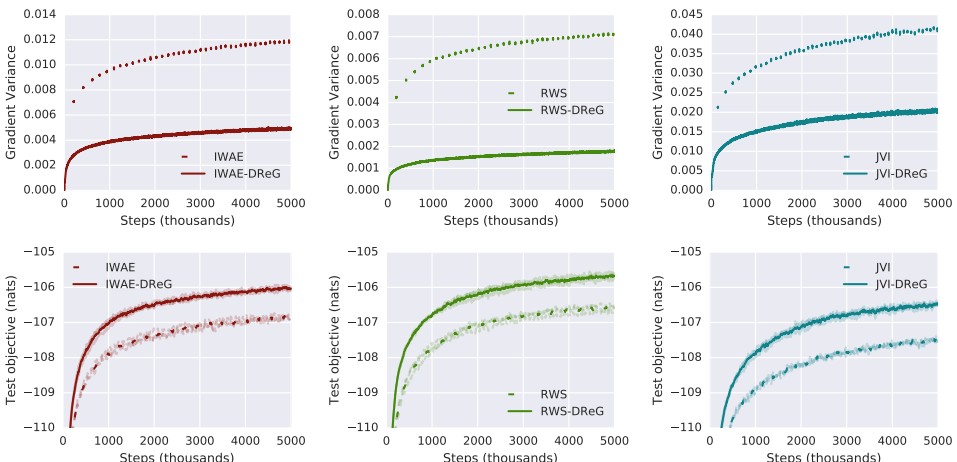

Figure 5: Omniglot generative modeling trained according to IWAE (left), RWS (middle), and JVI (right). The top row compares the variance of the original gradient estimator (dashed) with the variance of the doubly reparameterized gradient estimator (solid). The bottom row compares test performance. The left and middle plots show the IWAE (stochastic) lower bound on the test set. The right plot shows the JVI estimator (which is not a bound) on the test set. The bold lines are the average over three trials, and individual trials are displayed as semi-transparent). All methods used $K = 64$.

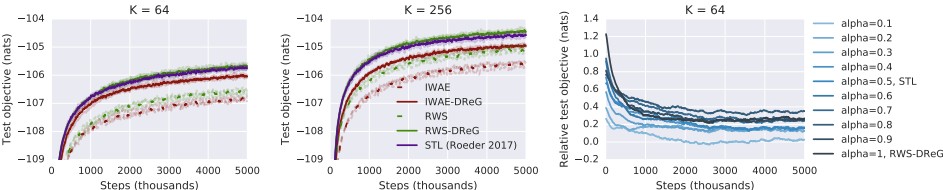

Figure 6: Log-likelihood lower bounds for structured prediction on Omniglot. The left plot uses $K = 64$ samples and the right plot uses $K = 256$ samples. For clarity the legend is shared between the plots. The bold lines are the average over three trials, and individual trials are displayed as semi-transparent). The right plot compares performance as the convex combination between IWAE-DReG and RWS-DReG is varied. To highlight differences, we plot the difference between the test IWAE bound and the test IWAE bound IWAE-DReG achieved at that step.

## 8.1 EQUIVALENCE BETWEEN REINFORCE GRADIENT AND REPARAMETERIZATION TRICK GRADIENT

Given a function $f(z, \phi)$, we have

$$\mathbb{E}_{q_\phi(z)}\left[f(z, \phi)\frac{\partial \log q_\phi(z)}{\partial \phi}\right] = \mathbb{E}_\epsilon\left[\frac{\partial f(z, \phi)}{\partial z}\frac{\partial z(\epsilon, \phi)}{\partial \phi}\right],$$

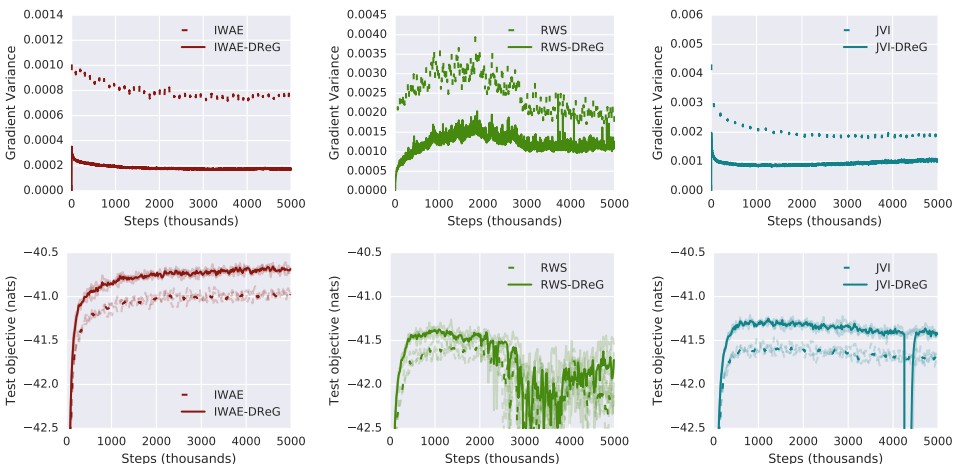

Figure 7: Structured prediction on MNIST according to IWAE (left), RWS (middle), and JVI (right). The top row compares the variance of the original gradient estimator (dashed) with the variance of the doubly reparameterized gradient estimator (solid). The bottom row compares test performance. The left and middle plots show the IWAE (stochastic) lower bound on the test set. The right plot shows the JVI estimator (which is not a bound) on the test set. The bold lines are the average over three trials, and individual trials are displayed as semi-transparent). All methods used $K = 64$.

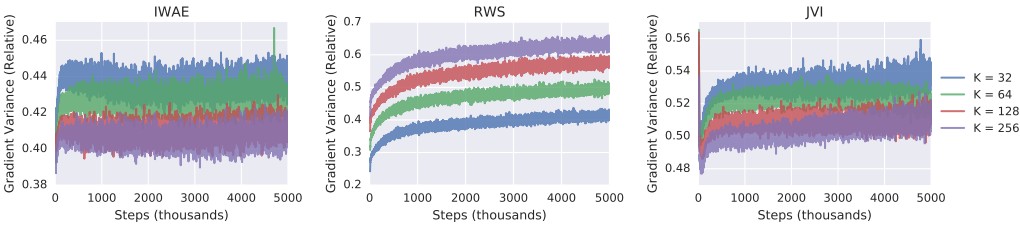

Figure 8: Variance of the gradient estimators on the MNIST generative modeling task. We plot the trace of the variance of the doubly reparameterized gradient estimator relative to the original gradient estimator for IWAE (left), RWS (middle), and JVI (right) as the number of samples (K) is varied.

for a reparameterizable distribution $q_\phi(z)$. To see this, note that

$$
\frac{d}{d\phi} \int_z q_\phi(z) f(z, \phi) \, dz = \int_z \frac{\partial}{\partial \phi} q_\phi(z) f(z, \phi) \, dz = \int_z f(z, \phi) \frac{\partial}{\partial \phi} q_\phi(z) + q_\phi(z) \frac{\partial}{\partial \phi} f(z, \phi) \, dz
$$

$$
= \int_z f(z, \phi) q_\phi(z) \frac{\partial \log q_\phi(z)}{\partial \phi} \, dz + \mathbb{E}_{q_\phi(z)} \left[ \frac{\partial f(z, \phi)}{\partial \phi} \right]
$$

$$
= \mathbb{E}_{q_\phi(z)} \left[ f(z, \phi) \frac{\partial \log q_\phi(z)}{\partial \phi} \right] + \mathbb{E}_{q_\phi(z)} \left[ \frac{\partial f(z, \phi)}{\partial \phi} \right],
$$

via the REINFORCE gradient. On the other hand,

$$
\frac{d}{d\phi} \int_z q_\phi(z) f(z, \phi) \, dz = \frac{d}{d\phi} \mathbb{E}_{q_\phi(z)} \left[ f(z, \phi) \right] = \frac{d}{d\phi} \mathbb{E}_\epsilon \left[ f(z(\epsilon, \phi), \phi) \right] = \mathbb{E}_\epsilon \left[ \frac{d}{d\phi} f(z(\epsilon, \phi), \phi) \right]
$$

$$
= \mathbb{E}_\epsilon \left[ \frac{\partial f(z, \phi)}{\partial z} \frac{\partial z(\epsilon, \phi)}{\partial \phi} \right] + \mathbb{E}_\epsilon \left[ \frac{\partial f(z, \phi)}{\partial \phi} \Big|_{z = z(\epsilon, \phi)} \right]
$$

$$
= \mathbb{E}_\epsilon \left[ \frac{\partial f(z, \phi)}{\partial z} \frac{\partial z(\epsilon, \phi)}{\partial \phi} \right] + \mathbb{E}_{q_\phi(z)} \left[ \frac{\partial f(z, \phi)}{\partial \phi} \right],
$$

via the reparameterization trick. Thus, we conclude that

$$\mathbb{E}_{q_\phi(z)}\left[f(z,\phi)\frac{\partial \log q_\phi(z)}{\partial \phi}\right] + \mathbb{E}_{q_\phi(z)}\left[\frac{\partial f(z,\phi)}{\partial \phi}\right] = \mathbb{E}_\epsilon\left[\frac{\partial f(z,\phi)}{\partial z}\frac{\partial z(\epsilon,\phi)}{\partial \phi}\right] + \mathbb{E}_{q_\phi(z)}\left[\frac{\partial f(z,\phi)}{\partial \phi}\right],$$

from which the identity follows.

## 8.2 ASYMPTOTIC ANALYSIS

At a high level, Rainforth et al. (2018) show that the expected value of the IWAE gradient of the inference network collapses to zero with rate $1/K$, while its standard deviation is only shrinking at a rate of $1/\sqrt{K}$. This is the essence of the problem that results in the SNR (expectation divided by standard deviation) of the inference network gradients going to zero at a rate $\mathcal{O}((1/K)/(1/\sqrt{K})) = \mathcal{O}(1/\sqrt{K})$, worsening with $K$. In contrast, Rainforth et al. (2018) show that the generation network gradients scales like $\mathcal{O}(\sqrt{K})$, improving with $K$.

Because the IWAE-DReG estimator is unbiased, we cannot hope to change the scaling of the expected value in $K$, but we can hope to change the scaling of the variance. In particular, in this subsection, we provide an informal argument, via the delta method, that the standard deviation of IWAE-DReG scales like $K^{-3/2}$, which results in an overall scaling of $\mathcal{O}(\sqrt{K})$ for the inference network gradient's SNR (i.e., increasing with $K$). Thus, the SNR of the IWAE-DReG estimator improves similarly in $K$ for both inference and generation networks.

We will appeal to the delta method on a two-variable function $g : \mathbb{R}^2 \to \mathbb{R}$. Define the following notation for the partials of $g$ evaluated at the mean of random variables $X, Y$,

$$g_x(X,Y) = \left.\frac{\partial g(x,y)}{\partial x}\right|_{(x,y)=(\mathbb{E}(X),\mathbb{E}(Y))}$$

The delta method approximation of $\mathrm{Var}(g(X,Y))$ is given by (Section 5.5 of Casella & Berger),

$$\mathrm{Var}(g(X,Y)) \approx g_x(X,Y)^2\mathrm{Var}(X) + 2g_x(X,Y)g_y(X,Y)\mathrm{Cov}(X,Y) + g_y(X,Y)^2\mathrm{Var}(Y)$$

Now, assume without loss of generality that $\phi$ is a single real-valued parameter. Let $u_i = w_i^2\frac{\partial \log w_i}{\partial z_i}\frac{\partial z_i}{\partial \phi}$, $X = \sum_{i=1}^K u_i$, and $Y = \sum_{i=1}^K w_i$. Let $g(X,Y) = X/Y^2$, then $g(X,Y)$ is the IWAE-DReG estimator whose variance we seek to understand. Letting $Z = \mathbb{E}(w_i)$ and $U = \mathbb{E}(u_i)$ we get in this case after cancellations,

$$\mathrm{Var}(g(X,Y)) \approx \frac{1}{Z^4}\frac{\mathrm{Var}(X)}{K^4} - \frac{4U}{Z^5}\frac{\mathrm{Cov}(X,Y)}{K^4} + \frac{4U^2}{Z^6}\frac{\mathrm{Var}(Y)}{K^4}$$

Because $w_i$ are all mutually independent, we get $\mathrm{Var}(Y) = K\mathrm{Var}(w_i)$. Similarly for $\mathrm{Var}(X)$ and $u_i$. Because the $w_i$ and $u_i$ are identically distributed and independent for $i \neq j$, we have $\mathrm{Cov}(X,Y) = K\mathrm{Cov}(w_i, u_i)$. All together we can see that $\mathrm{Var}(g(X,Y))$ scales like $K^{-3}$. Thus, the standard deviation scales like $K^{-3/2}$.

## 8.3 UNIFIED SURROGATE OBJECTIVES FOR ESTIMATORS

In the main text, we assumed that $\theta$ and $\phi$ were disjoint, however, it can be helpful to share parameters between $p$ and $q$ (e.g., (Fraccaro et al., 2016)). With the IWAE bound, we differentiate a single objective with respect to both the $p$ and $q$ parameters. Thus it is straightforward to adapt IWAE and IWAE-DReG to the shared parameter setting. In this section, we discuss how to deal with shared parameters in RWS.

Suppose that both $p$ and $q$ are parameterized by $\theta$. If we denote the unshared parameters of $q$ by $\phi$, then we can restrict the RWS wake update to only $\phi$. Alternatively, with a modified RWS wake update, we can derive a single surrogate objective for each scenario such that taking the gradient with respect to $\theta$ results in the proper update. For clarity, we introduce the following modifier notation for $p_\theta(x, z_i)$, $q_\theta(z_i|x)$, and $w_i$ which are functions of $\theta$ and $z_i = z(\theta, \epsilon_i)$. We use $\tilde{X}$ to mean $X$ with stopped gradients with respect to $z_i$, $\hat{X}$ to mean $X$ with stopped gradients with respect to $\theta$ (but not $\theta$ is not stopped in $z(\theta, \epsilon_i)$), and $\bar{X}$ to mean $X$ with stopped gradients for all variables. Then, we can use the following surrogate objectives:

IWAE:

$$L_{IWAE}(\theta) = \mathbb{E}_{\epsilon_{1:K}} \left[ \sum_{i=1}^{K} \frac{\bar{w}_i}{\sum_j \bar{w}_j} \log w_i \right] \tag{11}$$

DReG IWAE:

$$L_{DReG-IWAE}(\theta) = \mathbb{E}_{\epsilon_{1:K}} \left[ \sum_{i=1}^{K} \frac{\bar{w}_i}{\sum_j \bar{w}_j} \log \tilde{p}_\theta(x, z_i) + \left( \frac{\bar{w}_i}{\sum_j \bar{w}_j} \right)^2 \log \hat{w}_i \right] \tag{12}$$

RWS:

$$L_{RWS}(\theta) = \mathbb{E}_{\epsilon_{1:K}} \left[ \sum_{i=1}^{K} \frac{\bar{w}_i}{\sum_j \bar{w}_j} \left( \log \tilde{p}_\theta(x, z_i) + \log \tilde{q}_\theta(z_i|x) \right) \right] \tag{13}$$

DReG RWS:

$$L_{DReG-RWS}(\theta) = \mathbb{E}_{\epsilon_{1:K}} \left[ \sum_{i=1}^{K} \frac{\bar{w}_i}{\sum_j \bar{w}_j} \log \tilde{p}_\theta(x, z_i) + \left( \frac{\bar{w}_i}{\sum_j \bar{w}_j} - \left( \frac{\bar{w}_i}{\sum_j \bar{w}_j} \right)^2 \right) \log \hat{w}_i \right] \tag{14}$$

STL:

$$L_{STL}(\theta) = \mathbb{E}_{\epsilon_{1:K}} \left[ \sum_{i=1}^{K} \frac{\bar{w}_i}{\sum_j \bar{w}_j} \left( \log \tilde{p}_\theta(x, z_i) + \log \hat{w}_i \right) \right] \tag{15}$$

DReG($\alpha$):

$$L_{DReG(\alpha)}(\theta) = \mathbb{E}_{\epsilon_{1:K}} \left[ \sum_{i=1}^{K} \frac{\bar{w}_i}{\sum_j \bar{w}_j} \log \tilde{p}_\theta(x, z_i) + \left( \alpha \frac{\bar{w}_i}{\sum_j \bar{w}_j} + (1 - 2\alpha) \left( \frac{\bar{w}_i}{\sum_j \bar{w}_j} \right)^2 \right) \log \hat{w}_i \right] \tag{16}$$

The only subtle difference is that DReG($\alpha = 0.5$) does not correspond exactly to STL due to the scaling between terms:

$$L_{DReG(\alpha=0.5)}(\theta) = \mathbb{E}_{\epsilon_{1:K}} \left[ \sum_{i=1}^{K} \frac{\bar{w}_i}{\sum_j \bar{w}_j} \left( \log \tilde{p}_\theta(x, z_i) + 0.5 \log \hat{w}_i \right) \right] \tag{17}$$

