# OpenReview forum: "Doubly Reparameterized Gradient Estimators for Monte Carlo Objectives"
_ICLR.cc/2019/Conference_

### Official Review · AnonReviewer1 · 2018-10-24
**Proposed method is interesting but additional experiments are needed**

**Rating:** 6
**Confidence:** 4

**Review:**

Overall:
This paper works on improving the gradient estimator of the ELBO. Author experimentally found that the estimator of the existing work(STL) is biased and proposed to reduce the bias by using the technique like  REINFORCE.
The problem author focused on is unique and the solution is simple, experiments show that proposed method seems promising.

Clarity:
The paper is clearly written in the sense that the motivation of research is clear, the derivation of the proposed method is easy to understand.

Significance:
I think this kind of research makes the variational inference more useful, so this work is significant. But I cannot tell the proposed method is really useful, so I gave this score.
The reason I doubt the reason is that as I written in the below, the original STL can handle the mixture of Gaussians as the latent variable but the proposed method cannot. So I do not know which is better and whether I should use this method or use the original STL with flexible posterior distribution to tighten the evidence lower bound. I think additional experiments are needed. I know that motivation is a bit different for STL and proposed method but some comparisons are needed.

Question and minor comments:
In the original paper of STL, the author pointed out that by freezing the gradient of variational parameters to drop the score function term, we can utilize the flexible variational families like the mixture of Gaussians.
In this work, since we do not freeze the variational parameters, we cannot utilize the mixture of Gaussians as in the STL. IWAE improves the lower bound by increasing the samples, but we can also improve the bound by specifying the flexible posteriors like the mixture of Gaussians in STL.
Faced on this, I wonder which strategy is better to tighten the lower bound, should we use the STL with the mixture of Gaussians or use the proposed method?
To clarify the usefulness of this method, I think the additional experimental comparisons are needed.

About the motivation of the paper, I think it might be better to move the Fig.1 about the Bias to the introduction and clearly state that the author found that the STL is biased "experimentally".

Followings are minor comments.
In experiment 6.1, I'm not sure why the author present the result of K ELBO estimator in the plot of Bias and Variance.
I think author want to point that when K=1, STL is unbiased with respect to the 1 ELBO, but when k>1, it is biased with respect to IWAE estimator.
However, the objective of K ELBO and IWAE are different, it may be misleading. So this should be noted in the paper.

In Figure 3, the left figure, what each color means? Is the color assignment is the same with the middle figure?
(Same for Figure 4)

---

> ### Author Response · Authors · 2018-11-17
> **Author response**
>
> Recent work on reparameterizing mixture distributions has shown that the necessary gradients can be computed with the implicit reparameterization trick (Graves 2016, Jankowiak & Obermeyer 2018; Jankowiak & Karaletsos 2018; Figurnov et al. 2018).  Using this approach to reparameterize the mixture, DReGs readily apply when q is a Gaussian mixture model. We mention this explicitly in the text now.
>
> Eq. 6 explicitly characterizes the bias in STL. There is no reason to believe this term analytically vanishes, and we confirm numerically that it is non-zero in the toy Gaussian example. We believe this is sufficient to support our claim of bias.
>
> We present the K ELBO results in these plots to be consistent with previous work (Rainforth et al. 2018). We agree that it can be misleading for the reasons you indicated, so we now explicitly call this out in the maintext.
>
> Yes, the color assignment is the same. We note this in the caption for both figures now.

---

### Official Review · AnonReviewer2 · 2018-11-02
**Good paper**

**Rating:** 7
**Confidence:** 5

**Review:**

The paper observes the gradient of multiple objective such as IWAE, RWS, JVI are in the form of some “reward” multiplied with score function which can be calculated with one more reparameterization step to reduce the variance. The whole paper is written in a clean way and the method is effective.

I have following comments/questions:

1. The conclusion in Eq(5) is correct but the derivation in Sec. 8.1. may be arguable. Writing \phi and \tilde{\phi} at the first place sets the partial derivative of \tilde{\phi}  to \phi as 0. But the choice of \tilde{\phi} in the end is chosen as \phi. If plugging  \phi to \tilde{\phi}, the derivation will change. The better way may be calculating both the reparameterization and reinforce gradient without redefining a \tilde{\phi}.

2. How does the variance of gradient calculated where the gradient is a vector? And how does the SNR defined in the experiments?

3. How does the variance reduction from DReG changes with different value of K?

4. Is there any more detailed analysis or intuition why the right hand side of Eq(5) has lower variance than the left hand side?

---

> ### Author Response · Authors · 2018-11-16
> **Author response**
>
> Thank you for the helpful suggestions.
>
> 1. Thank you for pointing out this source of confusion. The correctness of the proof is related to the fact that \frac{\partial}{\partial \phi} g(\phi, \tilde{\phi}) |_{\tilde{\phi} = \phi} != \frac{\partial}{\partial \phi} g(\phi, \phi). On the left hand side the derivative is taken first, which results in a function of \phi and \tilde{\phi}, which we then evaluate. As you note, this is not equivalent to setting \tilde{\phi} = \phi, and then taking the derivative. We want the former. Following your suggestion, we have completely rewritten the proof to avoid this confusing step.
>
> 2. We used the trace of the Covariance matrix (normalized by the number of parameters) to summarize the variance, and we implemented this by maintaining exponential moving average statistics. SNR was computed as the mean of the estimator divided by the standard deviation (as in Rainforth et al. 2018). We added this information as footnotes in the maintext.
>
> 3. We have added a plot of the variance of the gradient estimator as K changes (Appendix Fig. 8). We found that as K increases, for IWAE and JVI, the variance of the doubly reparameterized gradient estimator slowly decreases relative to the variance of the original gradient estimator. On the other hand for RWS, we found that as K increases, the variance of the doubly reparameterized gradient estimator gradually increases relative to the variance of the original gradient estimator. However, we emphasize that in all cases, the variance of the doubly reparameterized gradient estimator was less than the variance of the original gradient estimator.
>
> 4. Yes, intuitively, the right hand side directly takes advantage of the gradient of f whereas the left hand side ends up computing something akin to finite differences. We have added a sentence explaining this intuition in the maintext.

---

### Official Review · AnonReviewer3 · 2018-11-04
**Doubly Reparameterized Gradient Estimators for Monte Carlo Objectives**

**Rating:** 7
**Confidence:** 3

**Review:**

This paper applies a reparameterization trick to estimate the gradients objectives encountered in variational autoencoder based frameworks with continuous latent variables.  Especially the authors use this double reparameterization trick on Importance Weighted Auto-Encoder (IWAE) and Reweighted Wake-Sleep (RWS)  methods. Compared to IWAE, the developed method's SNR does not go to zero with increasing the number of particles.

Overall, I think the idea is nice and the results are encouraging. I checked all the derivations, and they seem to be correct. Thus I recommend this paper to be accepted in its current form.

---

> ### Author Response · Authors · 2018-11-16
> **Author response**
>
> Thank you for checking the derivations. We appreciate the positive comments.

---

### Author Response · Authors · 2018-11-17
**Updated manuscript**

We have updated the manuscript based on reviewer feedback. Apart from clarifying edits, we have rewritten the derivation in Appendix 8.1 and included a plot of variance for several values of K as Appendix Figure 8.

---

> ### Author Response · Authors · 2018-12-26
> **Camera ready update**
>
> We have uploaded the final version with a link to the source code used for the experiments ( https://sites.google.com/view/dregs ).

---

### Meta-Review · Area_Chair1 · 2018-12-14
**An useful identity that helps improve existing training algorithms for deep generative models**

**Confidence:** 5
**Recommendation:** Accept (Poster)

**Metareview:**

The paper is well written and easy to follow. The experiments are adequate to justify the usefulness of an identity for improving existing multi-Monte-Carlo-sample based gradient estimators for deep generative models. The originality and significance are acceptable, as discussed below.

The proposed doubly reparameterized gradient estimators are built on an important identity shown in Equation (5). This identity appears straightforward to derive by applying both score-function gradient and reparameterization gradient to the same objective function, which is expressed as an expectation. The AC suspects that this identity might have already appeared in previous publications / implementations, though not being claimed as an important contribution / being explicitly discussed. While that identity may not be claimed as the original contribution of the paper if that suspicion is true, the paper makes another useful contribution in applying that identity to the right problem: improving three distinct training algorithms for deep generative models. The doubly reparameterized versions of IWAE and reweighted wake-sleep (RWS) further show how IWAE and RWS are related to each other and how they can be combined for potentially further improved performance.

The AC believes that the paper makes enough contributions by well presenting the identity in (5) and applying it to the right problems.

---

> ### Author Response · Authors · 2018-12-26
> **RE:**
>
> Thank you for pointing out this source of confusion. Indeed, the identity has been used extensively in previous publications and we are not claiming it as an original contribution of the paper. We introduce the identity in Eq. 5 as a "well-known equivalence".  As the AC notes, our contribution is in the application of the identity, the experimental evaluation, and the theoretical asymptotic analysis.